# Modified Protocol of Harvesting, Extraction, and Normalization Approaches for Gas Chromatography Mass Spectrometry-Based Metabolomics Analysis of Adherent Cells Grown Under High Fetal Calf Serum Conditions

**DOI:** 10.3390/metabo10010002

**Published:** 2019-12-18

**Authors:** Raphaela Fritsche-Guenther, Anna Bauer, Yoann Gloaguen, Mario Lorenz, Jennifer A. Kirwan

**Affiliations:** 1Berlin Institute of Health Metabolomics Platform, Berlin Institute of Health (BIH), 10178 Berlin, Germany; raphaela.fritsche@mdc-berlin.de (R.F.-G.); anna.bauer@mdc-berlin.de (A.B.); yoann.gloaguen@mdc-berlin.de (Y.G.); 2Max-Delbrück-Center for Molecular Medicine (MDC) in the Helmholtz Association, 13125 Berlin, Germany; 3Core Unit Bioinformatics, Berlin Institute of Health (BIH), 10178 Berlin, Germany; 4German Centre for Cardiovascular Research (DZHK), partner site Berlin, 13353 Berlin, Germany; mario.lorenz@charite.de; 5Charité-Universitätsmedizin Berlin, corporate member of Freie Universität Berlin, Humboldt-Universität zu Berlin, and Berlin Institute of Health, Medizinische Klinik für Kardiologie und Angiologie, Campus Mitte, 10117 Berlin, Germany

**Keywords:** GC-MS, 20% FCS, harvesting, extraction, metabolites, normalization

## Abstract

A gas chromatography mass spectrometry (GC-MS) metabolomics protocol was modified for quenching, harvesting, and extraction of metabolites from adherent cells grown under high (20%) fetal calf serum conditions. The reproducibility of using either 50% or 80% methanol for quenching of cells was compared for sample harvest. To investigate the efficiency and reproducibility of intracellular metabolite extraction, different volumes and ratios of chloroform were tested. Additionally, we compared the use of total protein amount versus cell mass as normalization parameters. We demonstrate that the method involving 50% methanol as quenching buffer followed by an extraction step using an equal ratio of methanol:chloroform:water (1:1:1, *v/v/v*) followed by the collection of 6 mL polar phase for GC-MS measurement was superior to the other methods tested. Especially for large sample sets, its comparative ease of measurement leads us to recommend normalization to protein amount for the investigation of intracellular metabolites of adherent human cells grown under high (or standard) fetal calf serum conditions. To avoid bias, care should be taken beforehand to ensure that the ratio of total protein to cell number are consistent among the groups tested. For this reason, it may not be suitable where culture conditions or cell types have very different protein outputs (e.g., hypoxia vs. normoxia). The full modified protocol is available in the Supplementary Materials.

## 1. Introduction

Reproducibility and robustness when performing metabolomics experiments are essential to guarantee that resulting biological information is both accurate and meaningful, and to reduce the likelihood of false discovery due to technical variation [1]. Cell cultures are often used as models for testing biological concepts. Therefore, an optimized and standardized protocol for efficient quenching, harvesting, and extraction of cells is required. There are many studies focusing on sample preparation methods, and many cell-dependent methods have been optimized for the application of cell metabolomics for analyzing metabolic fingerprints [2,3,4,5,6,7,8,9,10]. However, no preparation method is broadly applicable for all cell types or cultivation conditions. We had previously discovered that, in our hands, certain cell extraction methods were inappropriate for the cell types we were using, typically because they did not adequately separate the protein layer from the solvents. This was assumed to be a result of inappropriate volumes and ratios of the solvents to adequately extract the cell lipid content.

To prepare cells for metabolomics analyses, quenching at the point of harvesting aims to inactivate intracellular enzymes and stop metabolism to avoid degradation and alterations of sample composition. Sample preparation should be highly reproducible, robust, and fast to allow high-throughput studies [5]. As is known, metabolic reactions occur in milliseconds [11]. Therefore, it is necessary to quench metabolism quickly, most often by using ice-cold organic solvents. After the quenching and harvesting step, metabolite extraction is crucial and often the rate-limiting step. Extraction is the process by which specific compounds, or whole classes of compounds, e.g., polar metabolites are selectively separated from others (e.g., lipids, proteins) and is normally conducted to yield cleaner sample preparations for analysis. There has been much discussions over which extraction solvents are the best for quenching and measurement of metabolites [12]. Biphasic, liquid-liquid extraction is often used to extract metabolites, typically based on Folch or Bligh–Dyer methods [13,14,15]. Polar solutions like methanol (MeOH) in water (H_2_O) are often paired with nonpolar organic solvents, such as chloroform (CHCl_3_), to form a two-phase system. The chemical properties, volumes, and solvent ratios of the organic and aqueous solvents must be considered carefully. These parameters can significantly affect the extraction efficiency of metabolites and the experimental reproducibility.

Fetal calf serum (FCS) contains a large number of nutritional and macromolecular factors such as amino acids, sugars, lipids, and hormones essential for cell culture growth. For most cell lines, 10% FCS is used; however, some primary cells, such as human umbilical vein endothelial cells (HUVECs), need to be cultured in higher FCS conditions (e.g., medium with 20% FCS) [16,17,18,19,20,21,22]. To our knowledge, there have been no reports to date documenting the robustness and efficiency of harvesting and extraction protocols for metabolic profiling of adherent cells grown under high FCS conditions. Ideally, this would be undertaken using a full design-of-experiments approach investigating and fully testing multiple parameters to optimize the method. However, due to limitations in both resources and numbers of primary cell cultures, an incremental optimization approach can be taken to reach a predefined reproducibility and metabolite detection threshold. This approach enables an early exclusion of conditions that do not fit one’s criteria, therefore making the most out of the material available.

Therefore, the aim of this study was to compare and enhance our existing protocol for a reliable and reproducible harvesting and extraction of adherent cells grown in media supplemented with 20% FCS. As a comparison, standard cell culture conditions (10% FCS) were also investigated. All analyses were conducted by gas chromatography-mass spectrometry (GC-MS). The use of either 50% or 80% MeOH for quenching of cells were compared for sample harvest. In addition, two different ratios and three different final volumes of MeOH to CHCl_3_ to H_2_O were compared to investigate the efficiency and reproducibility of intracellular metabolite extraction. To achieve robust normalization of the samples, total protein amount versus absolute cell mass was tested.

## 2. Results

### 2.1. Preliminary Experiments on Phase Separation

A previous protocol used in our lab for adherent cells growing under 10% FCS was a modified Bligh–Dyer extraction to separate lipid and polar metabolites [23,24]. The protocol includes the use of 50% MeOH for cell quenching followed by an extraction using a 1:0.4:1, *v/v/v* MeOH:CHCl_3_:H_2_O ratio. Routinely, 3 mL of the polar phase is collected and used for further GC-MS measurement. For HUVECs cultured under conditions of 20% FCS, we discovered that this protocol was not sufficient to properly extract the cells. We discovered a fluffy protein pellet rather than the compacted, well demarcated protein pellet we normally achieve. This had consequences for the reproducibility of the measured total protein amount resulting in a relative standard deviation (RSD) for HUVECs of over 25% (European Medicine Agency (EMA) recommendation is RSD < 15%, http://www.ema.europa.eu/). To check this was not a cell-specific effect, the HCT116 colorectal carcinoma cell line was cultured and extracted in an identical way. Results were similar for the HCT116 cells with total protein RSDs also over 25% for high and standard FCS conditions, as seen in Table 1 and Figure 1 50_LOW. This suggests that extraction may not be complete and thus the technical reproducibility was larger than necessary. This prompted us to look at the entire process of harvesting and extraction.

In our study, four different extraction conditions were tested. The four conditions can be summarized here as: (1) 50_LOW: 50% MeOH quenching solution with 1 mL of CHCl_3_ for extraction with a final ratio of 1:0.4:1 (MeOH:CHCl_3_:H_2_O) and collection of 3 mL polar phase for analysis; (2) 50_MEDIUM: 50% MeOH quenching solution with 2.5 mL of CHCl_3_ for extraction with a final ratio of 1:1:1 (MeOH:CHCl_3_:H_2_O) and collection of 3 mL polar phase for analysis; (3) 80_HIGH: 80% MeOH quenching solution with 4 mL of CHCl_3_ for extraction with a final ratio of 1:1:1 (MeOH:CHCl_3_:H_2_O) and collection of 6 mL polar phase for analysis; and (4) 50_HIGH: 50% MeOH quenching solution with 4 mL of CHCl_3_ for extraction with a final ratio of 1:1:1 (MeOH:CHCl_3_:H_2_O) and collection of 6 mL polar phase for analysis.

### 2.2. Experiment 1: Determining the Optimum Concentration of MeOH as a Quenching and Extraction Solution for Cell Harvesting

The quenching solution used is important as it has a dual function of being the first extraction step and being required to quench metabolism quickly. Many reports suggest 80% MeOH for metabolite quenching and extraction [7,25,26]; our current protocol uses 50% MeOH (50_LOW). Therefore, both 50% and 80% MeOH solutions were tested as quenching and extraction buffers for harvesting of adherent cells cultured in 10% and 20% FCS. The modified Bligh–Dyer extraction step which follows the quenching step will influence the results. In a first test, we used the same volume of 1 mL CHCl_3_ for the extraction step (as used in our current protocol) and observed that for 80% MeOH, no phase separation occurred, as seen in Appendix A. This is assumed to be due to the relative ratios of CHCl_3_ and to polar solvents. Increasing the ratio of MeOH:CHCl_3_:H_2_O to 1:1:1 *v/v/v* resulted in a phase separation. In order to keep the ratio of the final solvent composition constant, the extracted volumes of H_2_O and MeOH needed to be adjusted accordingly to account for whether 50% or 80% MeOH was used in the first extraction step.

The performance of individual quenching solvents was determined by a range of measures including (i) number of detected metabolites, (ii) median RSD of all detected metabolites (a measure of reproducibility), (iii) the percentage of metabolites with an individual RSD < 30% (a generally accepted tolerance limit for GC-MS metabolomics for any individual metabolite [27]), and (iv) the sum of normalized peak area for all annotated metabolites (henceforth referred to as sum of area), a proxy measure of the overall sensitivity of the method.

In the HCT116 cells, the conditions were equivalent in terms of number of metabolites detected, as seen in Appendix A, except for the detection of fructose-6-phosphate, when 80% MeOH was used with 10% culture conditions, as seen in Appendix A, comparing 50_HIGH and 80_HIGH. For the HUVECs, a lower number of metabolites was found, as seen in Appendix A and Appendix A; 34 metabolites for 50_HIGH and 36 for 80_HIGH. We assume this lower number of metabolites is due to the generally lower average cell count achieved by the HUVECs (mean cell counts: male HUVECs: 1.5 × 10^6^ cells; female HUVECs: 3 × 10^6^ cells; HCT116: 6 × 10^6^ cells). The metabolites detected also had some very minor differences (e.g., proline was only detected in 50_HIGH and tyrosine, uracil, and ribose-5-phosphate were only detected in 80_HIGH).

The reproducibility of both quenching solvents was then assessed using RSDs as an indicator. For HCT116 cells and HUVECs in both culture conditions, 50% MeOH performed better overall as a quenching solvent compared to 80% MeOH, comparing 50_HIGH and 80_HIGH in Table 2, Table 3, Figure 2, Appendix A. In HCT116 cells 50_HIGH has a lower median RSD compared to 80_HIGH (14% compared to 23%, and 26% compared to 30%, for 10% FCS and 20% FCS, respectively). In addition, there was a higher percentage of individual metabolites with RSD < 30% in the 50_HIGH (79% compared to 67%, and 73% compared to 53%, for 10% FCS and 20% FCS, respectively). In line, the sum of the area was higher in 50_HIGH compared to 80_HIGH for both FCS culture conditions (Figure 3; 74 compared to 56, and 35 compared to 26, for 10% FCS and 20% FCS, respectively). In HUVECs, the median RSD was lower in 50_HIGH (13%) compared to 80_HIGH conditions (36%), with a higher percentage of individual metabolite with RSDs < 30% (74% 50_HIGH compared to 44% 80_HIGH). The measured sum of area for HUVECs was also better for 50_HIGH (3.8 compared to 3.2) According to these results, 50% MeOH was deemed to be a better quenching solvent for both types of cells. However, for particular metabolites, 80% MeOH may be more appropriate.

### 2.3. Experiment 2: Determination of the Optimal Volume and Ratio of MeOH:CHCl_3_:H_2_O for a Metabolite Extraction Solution

The ratio of MeOH:CHCl_3_:H_2_O is important for: (i) the efficiency and reproducibility of extraction; (ii) a clear phase separation; and (iii) a robust method for protein collection for normalization or for concurrent proteomics (see Section 2.1). Different ratios, final volumes, and polar phase volume collection for extraction were evaluated, as seen in Table 4. This step was carried out before 50_HIGH was deemed to be optimum for quenching, but, for convenience, is presented here in the same order in which it appears in the standard operating protocol. As the different extraction methods led to different final volumes, leading to different theoretical concentrations, more volume was analyzed in some samples to normalize and compensate for this concentration difference.

The evaluation of the different ratios and volumes revealed no significant changes in either the number of detected metabolites, as seen in Table 5, or particular metabolite classes independent of the analyzed cells and growth conditions, as seen in Appendix A.

In HCT116 cells, the use of 50_MEDIUM compared to 50_LOW led to lower RSDs (20% compared to 51%) for total protein measurement in cells grown under 10% FCS. HCT116 cells cultured with 20% FCS showed increased RSDs (42% compared to 30%) in the 50_MEDIUM condition, as seen in Table 1. When using 80_HIGH conditions, the pellet was more compact, as seen in Figure 1, and easier to extract, which is reflected in a low RSD (13% for 10% FCS and 8% for 20% FCS) of the measured protein amount in HCT116 cells. When using 50_HIGH conditions, a compact protein pellet could be observed, leading to a more efficient protein extraction, and resulting in RSDs (14% for 10% FCS and 16% for 20% FCS) similar to 80_HIGH conditions. For HUVECs, the RSD of the measured total protein was reduced when using 80_HIGH compared to 50_MEDIUM and 50_HIGH conditions, as seen in Appendix A. 50_MEDIUM compared to 50_LOW conditions showed a lower RSD (15% compared to 49%, respectively).

The reproducibility of GC-MS analysis as measured by the median RSD of the individual metabolites or combined as biological pathways, and the percentage of metabolites with RSD < 30%, revealed the best results when using 50_HIGH conditions for HCT116 cells, as seen in Table 2, Table 3 and Figure 2. Likewise, in HUVECs, the best results were shown using 50_HIGH conditions, as seen in Appendix A, Appendix A and Appendix A.

Independent of the FCS setting in HCT116 cells, the highest sum of area was found using 50_HIGH conditions, as seen in Figure 3. For HUVECs, the highest sum of area was found in cells in the 50_MEDIUM condition; however, the variability was high, as seen in Appendix A. Comparing 80_HIGH and 50_HIGH conditions measured as a separate batch, a higher sum of area was found using 50% MeOH as quenching buffer.

In summary, our results indicate that using an extraction ratio of 1:1:1, *v/v/v* MeOH:CHCl_3_:H_2_O and a polar phase volume of 6 mL achieved good results for samples used for both GC-MS measurement and protein extraction for cells grown under 10% and 20% FCS. Lower individual and combined RSDs and higher peak areas could be detected. The direct comparison of 50% or 80% MeOH as quenching buffer showed a higher sum of area and lower median RSDs when using 50_HIGH conditions for the analyzed cells and FCS conditions. The number of annotated metabolites was equal across all methods. The full modified protocol is available in the Appendix A.

### 2.4. Experiment 3: Total Protein versus Cell Mass as a Normalization Strategy

Adherent cells need to be detached from the bottom of the cell culture flask for further metabolomics analyses. Trypsinization is known to lead to alteration of metabolism, so we employed direct quenching of cells with physical scraping as an alternative [5,28]. However, this complicates the utilization of the cell count as a normalization parameter. Therefore, we compared the use of total protein amount versus the cell mass as a normalization strategy.

After normalizing to total protein, HCT116 cells grown under 10% FCS showed a slightly lower median RSD of 24% for individual metabolite detection compared to normalization with cell mass (median RSD 28%), as seen in Table 6. For HCT116 cells cultivated in 20% FCS, a median RSD of 35% and 25%, respectively, was found if normalized to total protein or cell mass. For HUVECs, the median RSD was lower when using protein (52%) for normalization compared to cell mass (61%), as seen in Appendix A. The percentage of individual metabolite RSDs < 30% threshold was higher when using total protein (26%) compared to cell mass (18%) for normalization. Judging from Table 6 and Appendix A, neither of the normalization methods was superior. However, total protein was considerably easier to measure.

## 3. Discussion

Metabolomics has long been used for the analysis of human body fluids for clinical indications [29,30]. More recently, it is also being applied to cells or tissue [30]. Metabolite profiling of adherent growing mammalian cells is challenging, particularly due to the special requirements with respect to the sampling procedure. While harvesting of suspension cultures can generally be achieved through rapid centrifugation, the adherent cells first need to be detached from the bottom of the cell culture flask [30]. With a large number of metabolites to be analyzed simultaneously, it becomes difficult to find a single optimal extraction method. In this work, we set out to solve a particular challenge, which was the visible suboptimal extraction of cells grown under high (20%) FCS conditions. We focused on optimizing a liquid-liquid extraction protocol for the harvest and extraction of cells grown both under high (20%) or normal (10%) FCS conditions to enable reproducible and robust analysis of the detected metabolites. We compared our existing protocol using 50% MeOH to 80% MeOH as a quenching buffer, followed by an extraction using different ratios and volumes of CHCl_3_ and polar phases. Environmental and personal safety should also be considered and has resulted in a move away from CHCl_3_ towards other nonorganic solvents such as methyl-tert-butylether (MTBE) in recent years [31]. MTBE is both highly volatile and floats on the surface of the resulting biphasic mixture. As we were interested in analyzing only the polar metabolites in this study, it was excluded from the test solvents. We also looked at whether cell mass or total protein represented a better normalization strategy, since cell count cannot easily be used for adherent cells harvested for a metabolomics approach [32].

Direct quenching of the metabolism in adherent cells needs extraction with organic solvents such as MeOH. Various reports on the best solvent composition for metabolite extraction suggests 80% MeOH in H_2_O [7,26]; however, 50% MeOH is also used for metabolite extraction of adherent cells grown under 10% FCS [23]. We tried using both 50% MeOH and 80% MeOH as quenching buffers, followed by an extraction using different ratios and volumes of CHCl_3_ and polar phases in cells cultured with 20% FCS compared to standard cell culture conditions. Using the 50_HIGH method (50% MeOH followed by extraction using a 1:1:1, *v/v/v* MeOH:CHCl_3_:H_2_O ratio and collection of 6 mL polar phase) was considered to be the best of the extraction methods we tested. Using a higher ratio of CHCl_3_ and higher extraction solvent volumes gives a better phase separation and results in a dense protein pellet allowing a robust normalization. The percentage of highly polar solutions in the final ratio is also likely to be important for efficient extraction of polar metabolites and is reflected in the quenching results. The results may be different if nonpolar metabolites are also considered.

Normalization methods for cell culture are often problematic. Normalization is used to eliminate inter-run variability and biological variation [25]. Cell counts are inaccurate and are often performed on separate culture plates to the ones that are analyzed [33]. This is labor-intensive, impractical, and inefficient, especially with regard to large studies. Using cell count for normalization requires cell detachment using trypsin solutions, which can lead to metabolite leakage and changes in the metabolic pattern [5]. Total protein or cell mass has the advantage over cell counts that they can be performed using the same plate as that for the metabolomics analysis. One difficulty in using cell mass for normalization is the removal of the quenching buffer after centrifugation. Remnants of the quenching solution left in the vials will affect the final recorded cell mass, leading to more technical error. Further, the process of weighing the samples is technically laborious. Metabolites are contained in the buffer, meaning that it needs to be first removed for the weighing process and then re-added. This all needs to take place as quickly as possible, while keeping the sample as cold as possible so that it has minimal effects on reproducibility. As total protein can be measured following extraction, it can be done with more ease, although it has the disadvantage that samples cannot then be extracted according to their mass. They can, however, be reconstituted to equivalent concentrations before being analyzed if a drying down step is employed immediately after extraction. We have shown in the results that the extraction of the protein was reproducible enough to be considered an effective normalization method when the 50_HIGH method was used. For large numbers of samples, protein extraction is easier since the critical step of weighing and cooling is missing. Therefore, we recommend using total protein as a potential normalization strategy. We are aware of certain cell culture conditions that reduce the protein amount per cell. For this reason, full consideration should be given to whether the total protein is likely to be a good reflection of cell count or cell volume before it is used. In our hands, we have seen certain culture conditions, e.g., hypoxia, where cell protein production is not equivalent in the same cell type as in normoxia. In these circumstances, cell protein would probably not be recommended as a suitable normalization strategy.

In this study, we have tested and improved our existing method to enhance our reproducibility when analyzing adherent mammalian cells cultured under 20% FCS. For a truly optimized method, a design of experiments (DoE) approach is recommended. However, due to the difficulties of working with large numbers of cell cultures per batch, the cost of repeated experiments and the statistical understanding required to pursue a DoE approach, we instead chose to trial and optimize our existing method. We are aware that this incremental optimization approach may not result in the truly optimal method; however, it enables a faster development that limits material waste. The method is deemed to be optimal once certain reproducibility and metabolite detection criteria are reached. This represents a more realistic scenario for the thousands of labs required to run samples each year without the time and resources to fully optimize methods for each sample type.

In summary, the final modified 50_HIGH quenching, extraction, and normalization method (50% MeOH for harvest followed by extraction using a 1:1:1, *v/v/v* MeOH:CHCl_3_:H_2_O ratio and collection of 6 mL polar phase) can be recommended for the investigation of intracellular metabolites from adherent human cells grown under standard (10%) or high (20%) FCS conditions using a GC-MS platform.

## 4. Materials and Methods

### 4.1. Cell Culture

Two different human cell types were used in this study. We were interested mainly in a robust analysis of HUVEC male and female twin pair cells, but since these are human derived and therefore in limited supply, HCT116 cells were used to enable more replicates to be tested. For the same reason, a mix of male and female HUVECs were used.

The colorectal carcinoma cell line HCT116 was obtained from ATCC (American Type Culture Collection, Teddington, UK). The HCT116 cell line was maintained in DMEM (Dulbecco’s Modified Eagle Medium, Thermo Fischer Scientific, Waltham, MA, USA) supplemented with 10% or 20% FCS (Thermo Fischer Scientific, Waltham, MA, USA), 1% penicillin/streptomycin (Thermo Fischer Scientific, Waltham, MA, USA), 2 mM glutamine (Thermo Fischer Scientific, Waltham, MA, USA) and 1 g/L glucose (Sigma-Aldrich, St. Louis, MO, USA). HUVECs were isolated as previously described [17]. All cells were incubated in a humidified atmosphere of 5% CO_2_ in air at 37 °C.

### 4.2. Experiment 1: Determination of the Optimum Concentration of Methanol as a Quenching and Extraction Solution for Cell Harvesting

Cells were rapidly washed (20 s) with washing buffer (140 mM NaCl, 5 mM HEPES, pH 7.4, 37 °C) before they were quenched by ice-cold MeOH solution. Two solutions were tested: either 5 mL of 50% or 80% MeOH in H_2_O with a final concentration of 2 µg/mL cinnamic acid (for use as an internal standard). Immediately as the MeOH solution was added to the culture plate, cells were scraped into the methanol solution and the methanolic lysates were collected. The extracts were agitated to complete cell lysis and centrifuged to separate the layers (see Section 4.3).

### 4.3. Experiment 2: Determination of the Optimal Volume and Ratio of MeOH:CHCl_3_:H_2_O for a Metabolite Extraction Solution

After cell harvest, CHCl_3_ was added to the methanolic cell extracts, shaken for 60 min at 4 °C, and centrifuged at 4149× *g* for 10 min at 4 °C to separate the phases. Both the final ratio of MeOH:CHCl_3_:H_2_O and the final total volume was tested. Test conditions are summarized in Table 4.

The polar phase was collected and dried overnight at 30 °C at a speed of 1550× *g* at 0.1 mbar using a rotational vacuum concentrator (RVC 2-33 CDplus, Christ, Osterode am Harz, Germany). Samples were pooled after extraction and used as a quality control (QC) sample to test the technical variability of the instrument. QC samples were prepared alongside the samples in the same way. To generate backup samples, the dried polar phases were resuspended and split into two aliquots.

To further test whether the results of changing the ratios of MeOH:CHCl_3_:H_2_O are an effect of the ratio or of the total volume, we investigated changing the ratios while maintaining the volume. For the harvesting condition of 50% MeOH, a final volume of 6 mL was tested, resulting in 3 mL of polar phase that could be collected for analysis. For the harvesting condition using 80% MeOH, a final extraction volume of 12 mL was tested with 6 mL polar phase collected and dried for analysis. To perform a direct comparison of the same volume we used 5 mL of 50% MeOH for cell quenching followed by increased extraction volumes (4 mL CHCl_3_, additional 1.5 mL MeOH and 1.5 mL H_2_O) leading to a 1:1:1, *v/v/v* ratio of MeOH:CHCl_3_:H_2_O. With the increased volumes, 6 mL of polar phase could be collected and used for GC-MS measurement. The 80_LOW condition (5 mL 80% MeOH for quenching and 1 mL of CHCl_3_ for extraction) was discarded from the optimization tests since no phase separation could be achieved, as seen in Appendix A.

### 4.4. Experiment 3: Total Protein versus Cell Mass as a Normalization Strategy

To measure cell mass, at the point immediately following quenching and harvest of the cells, the quenched cells (50% MeOH) were centrifuged for 10 min at 10,000× *g* at 4 °C and the supernatant was carefully removed and collected into a new 15 mL Falcon tube stored on ice. The cell pellet, also kept on ice, was immediately weighed as fast as possible and the collected supernatant was added back to it, ready for metabolite extraction. The metabolites were extracted using a high volume of CHCl_3_ (4 mL) as determined in the previous section. After collection of the polar phases (6 mL), proteins were precipitated for each sample by addition of 8 mL 100% MeOH to avoid phase separation followed by centrifugation at 16,000× *g* speed for 10 min. The supernatant was carefully discarded. The pellet was air dried at room temperature and used for protein lysis and protein determination as described in Section 4.5.

### 4.5. Protein Extraction and Determination

For measurement of total protein amount, the pellet was resuspended in 8 M urea buffer (in 50 mM HEPES, pH 8.5). The protein concentration was determined using a bicinchoninic acid (BCA) assay (Thermo Fischer Scientific, Waltham, MA, USA) following the manufacturer’s instructions. In brief, 2 µL from each protein lysate was added to 2 µL reagent A, followed by the addition of 100 µL reagent B. After 30 min, absorbance was read at 562 nm using an Infinite M200Pro plate reader (TECAN, Männedorf, Switzerland). In order to calculate a calibration curve, a dilution series of bovine serum albumin (BSA, 1 mg/mL to 0.05 mg/mL) was included in the measurement.

### 4.6. GC-MS Metabolomics Measurement of Key Central Carbon Pathway Metabolites

All polar cell extracts were stored dry at −80 °C until analysis. Extracts were removed from the freezer and further dried in a rotational vacuum concentrator for 60 min before further processing to ensure there was no residual water which may influence derivatization efficiency. Dried cell extracts were dissolved in 15 µL of methoxyamine hydrochloride solution (40 mg/mL in pyridine) and incubated for 90 min at 30 °C with constant shaking, followed by the addition of 50 µL of *N*-methyl-N-[trimethylsilyl]trifluoroacetamide (MSTFA) and incubation at 37 °C for 60 min. The extracts were centrifuged for 10 min at 10,000× *g*, and aliquots of 25 µL were transferred into glass vials for GC-MS measurement. An identification mixture for reliable compound identification was prepared and derivatized in the same way and an alkane mixture for reliable retention index calculation was included [23]. Metabolite analysis was performed on a Pegasus 4D GC×GC TOF-MS-System (LECO Corporation, St. Joseph, MN, USA) complemented with an auto-sampler (Gerstel MPS DualHead with CAS4 injector, Mühlheim an der Ruhr, Germany). The samples were injected in split mode (split 1:5, injection volume 1 µL) in a temperature-controlled injector with a baffled glass liner (Gerstel, Mühlheim an der Ruhr, Germany). The following temperature program was applied during sample injection: for 2 min the column was allowed to equilibrate at 68 °C, a first temperature gradient was started with a rate of increase of 5 °C/min until a maximum of 120 °C was reached. Subsequently, the temperature gradient was changed such that the rate of increase was 7 °C/min up to a maximum temperature of 200 °C. This was increased to a 12 °C/min gradient up to a maximum temperature of 320 °C which was then held for 7.5 min. Gas chromatographic separation was performed on an Agilent 7890 (Agilent Technologies, Santa Clara, CA, USA), equipped with a VF-5ms column (Agilent Technologies, Santa Clara, CA, USA) of 30 m length, 250 µm inner diameter, and 0.25 µm film thickness. Helium was used as the carrier gas with a flow rate of 1.2 mL/min. The spectra were recorded in a mass range of 60 to 600 m/z with 10 spectra/s. The GC-MS chromatograms were processed with ChromaTOF software (LECO Corporation, St. Joseph, MN, USA) including baseline assessment, peak picking, and computation of the area.

### 4.7. Data Analysis

An in-house-created library and reference search including 45 most relevant metabolites from the central carbon metabolism (CCM) were used, as seen in Appendix A. The data were exported and merged by an in-house R script. The metabolites were considered valid when they appeared in a minimum of three out of five biological replicates (BR) in HCT116, two or three in female HUVECs, and two BR for male HUVECs. Male and female HUVECs were analyzed separately because of the availability of the material. The peak area of each metabolite was calculated by normalization to the internal standard, cinnamic acid, and additionally to the protein content (or cell mass). The performance of the different MeOH percentages were evaluated according to the following criteria: (1) number of detected metabolites, (2) relative peak area, and (3) reproducibility. The technical variation of the GC-MS run is shown in Appendix A.

## Figures and Tables

**Figure 1 metabolites-10-00002-f001:**
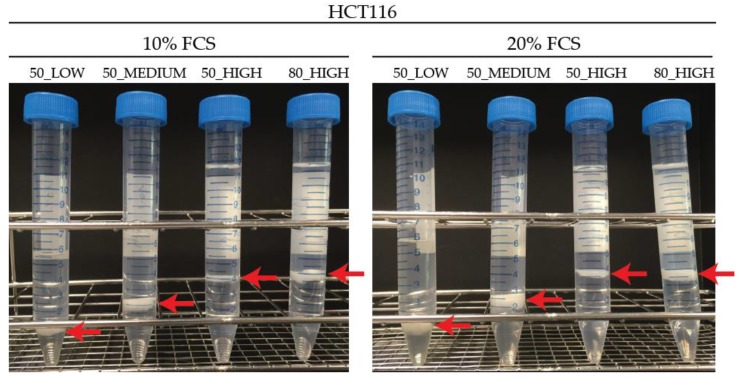
Extraction of HCT116 cells cultured in 10% and 20% fetal calf serum (FCS) conditions using different quenching solvents, extraction ratios, and polar volumes (protein layer shown by extra arrow on picture).

**Figure 2 metabolites-10-00002-f002:**
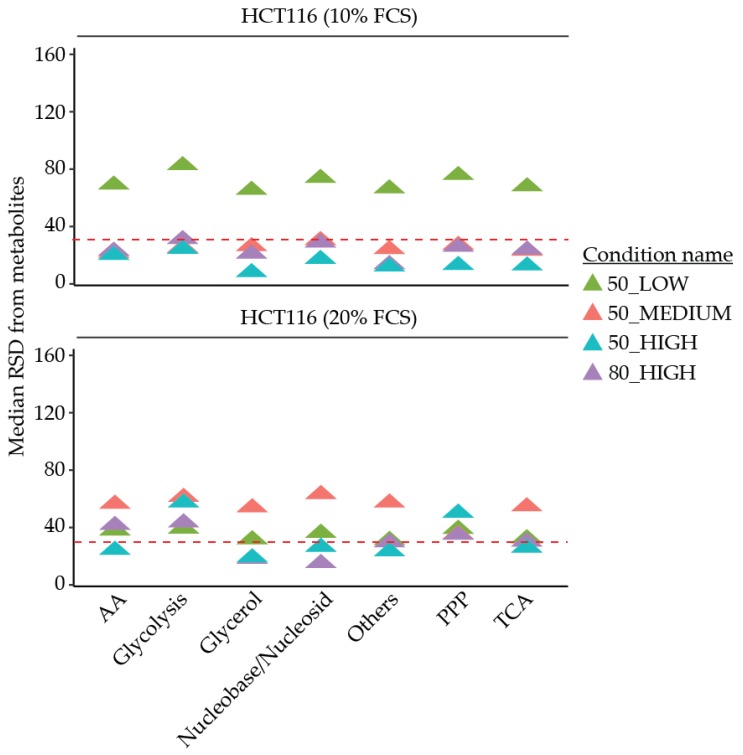
Median relative standard deviation (RSD) of individual metabolites separated by biological metabolite pathways of HCT116 cells cultured in 10% and 20% FCS conditions using different quenching solvents, extraction ratios, and polar volumes. The dashed line represents the maximum 30% RSD threshold advised by the Federal Drug Administration (FDA). AA: Amino acids. PPP: Pentose phosphate pathway. TCA: Tricarboxylic acid cycle.

**Figure 3 metabolites-10-00002-f003:**
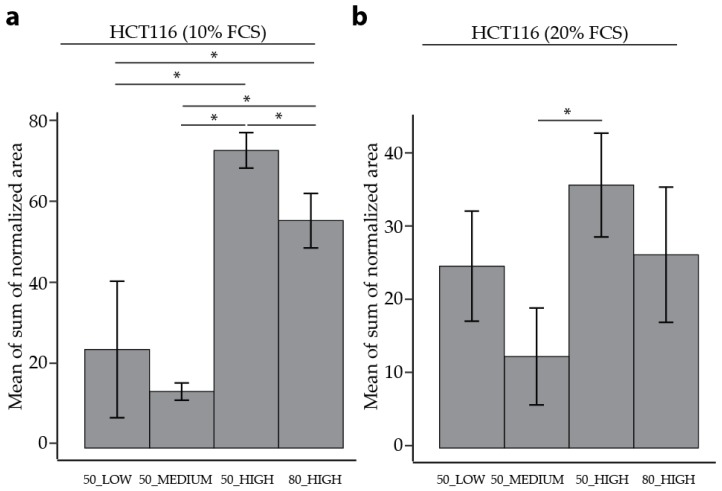
Comparison of the mean of sum of normalized peak area for different quenching and extraction methods. Data from a minimum three out of five HCT116 cells cultured in (**a**) 10% and (**b**) 20% FCS. The error bars represent the standard deviation of the biological replicates. Samples were analyzed using an unpaired Student’s *t-*test with *p* < 0.05 deemed as statistically significant.

**Table 1 metabolites-10-00002-t001:** Relative standard deviation (RSD) of measured protein amount of HCT116 cells cultured in 10% and 20% FCS conditions using different quenching solvents, extraction ratios, and polar volumes.

Condition Name	50_LOW	50_MEDIUM	50_HIGH	80_HIGH
Biological replicates 10% FCS	4	3	4	5
Biological replicates 20% FCS	5	3	4	4
HCT116 (10% FCS)	51	20	14	13
HCT116 (20% FCS)	30	42	16	8

**Table 2 metabolites-10-00002-t002:** Median relative standard deviation (RSD) per metabolite of HCT116 cells cultured in 10% and 20% FCS conditions using different quenching solvents, extraction ratios, and polar volumes.

Condition Name	50_LOW	50_MEDIUM	50_HIGH	80_HIGH
Biological replicates 10% FCS	4	3	4	5
Biological replicates 20% FCS	5	3	4	4
HCT116 (10% FCS)	68%	22%	14%	23%
HCT116 (20% FCS)	33%	55%	26%	30%

**Table 3 metabolites-10-00002-t003:** Percentage of metabolites with a relative standard deviation (RSD) < 30% of HCT116 cells cultured in 10% and 20% FCS conditions using different quenching solvents, extraction ratios, and polar volumes.

Condition Name	50_LOW	50_MEDIUM	50_HIGH	80_HIGH
Biological replicates 10% FCS	4	3	4	5
Biological replicates 20% FCS	5	3	4	4
HCT116 (10% FCS)	0%	73%	79%	67%
HCT116 (20% FCS)	15%	3%	73%	53%

**Table 4 metabolites-10-00002-t004:** Experimental set up for the conditions analyzed in the study.

Condition Name	50_LOW ( 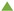 )	50_MEDIUM ( 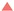 )	50_HIGH ( 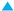 )	80_HIGH ( 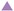 )
5 mL of MeOH harvest	50% (2.5 + 2.5) *	50% (2.5 + 2.5) *	50% (2.5 + 2.5) *	80% (4.0 + 1.0) *
MeOH [mL] for extraction	-	-	1.5	-
H_2_O [mL] for extraction	-	-	1.5	3.0
CHCl_3_ [mL] for extraction	1.0	2.5	4.0	4.0
Final volume [mL]	6.0	7.5	12	12
Final ratio *v/v/v*	1:0.4:1	1:1:1	1:1:1	1:1:1
Final polar volume [mL]	5.0	5.0	8.0	8.0
Used polar volume [mL]	3.0	3.0	6.0	6.0

*: mL of MeOH + H_2_O.

**Table 5 metabolites-10-00002-t005:** Number of annotated central carbon metabolites of HCT116 cells cultured in 10% and 20% FCS conditions using different quenching solvents, extraction ratios, and polar volumes.

Condition Name	50_LOW	50_MEDIUM	50_HIGH	80_HIGH
Biological replicates 10% FCS	4	3	4	5
Biological replicates 20% FCS	5	3	4	4
HCT116 (10% FCS)	38/45	39/45	38/45	39/45
HCT116 (20% FCS)	41/45	41/45	40/45	40/45

**Table 6 metabolites-10-00002-t006:** Median relative standard deviation (RSD) and percentage of metabolites with a RSD < 30% from four replicates of HCT116 cells cultured in 10% and 20% FCS conditions.

	HCT116 (10% FCS)		HCT116 (20% FCS)	
	Cell Mass	Protein	Cell Mass	Protein
Median RSD [%]	28	24	25	35
RSD < 30% [%]	55	60	60	33

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
