# Peer review of "Modified Protocol of Harvesting, Extraction, and Normalization Approaches for Gas Chromatography Mass Spectrometry-Based Metabolomics Analysis of Adherent Cells Grown Under High Fetal Calf Serum Conditions"

_metabolites, 2019, doi:10.3390/metabo10010002_

Round 1

Reviewer 1 Report

This paper presents a nice study of relevance to researchers working with adherent cell cultures, especially relevant to those working with cells in high FCS.  Even if the system being studied by other researchers is not the same as presented here, the approach taken by the authors represents a good model for how to evaluate variations on conditions to optimise quenching and extraction of cell metabolites.  The results are comprehensively and effectively presented and the authors have devised recommendations supported by their findings.  There are a fair few language usage errors (although mostly minor) that need to be addressed before publication.  There was one apparently errant reference (to a chapter that doesn't exist since this was not a contribution to a book, although perhaps the authors also prepared this paper for a book or thesis chapter).

Errors to be corrected are highlighted in the attached pdf.

Author Response

Response to Reviewer 1 Comments

We thank the reviewer for their constructive comments.

All changes were made according to reviewer’s suggestions.

Has this been copied from another publication? There should be no reference to chapters in this paper.

Comment to “see chapter 2.3”: Chapter was changed to section 2.3.

Reviewer 2 Report

The manuscript by Fritsche-Guenther et al. describes the optimization of a sample preparation protocol for GC-MS analysis of adherent cells that need to be grown at 20 % FCS. Authors also compare normalization using cell mass vs protein content. Final conditions selected include 50 % of methanol as quenching buffer, extraction with a methanol:chloroform:water mixture in a 1:1:1 proportion and collecting 6 mL of the polar phase and using protein content for further normalization. Authors provide the SOP at the end of supplementary material. My opinion is that the topic discussed here is of relevance. However, there are some parts of the manuscript that remain unclear and need to be improved. Special care should be taken on how the results are presented. My comments are below:

Major concerns:

-The way the authors present the tables is very redundant. I recommend clearly defining at the very beginning what the four conditions are, this is, explain the composition of each of 50_LOW, 50_MEDIUM, 50_HIGH and 80_HIGH and avoid having this chunk of text repeated all over in every single table and even figures (See Fig 1). Also make references to the name of each condition when comparisons are performed, instead of using terms like “comparing column 2 and 4” or sometimes as “comparing lane 2 and 4” (see Line 156).

-Section 2.2 is very fuzzy. Authors want to compare 50 and 80 % of methanol as quenching solution but the final volume of each condition is different, 7.5 and 12 mL, respectively. This affects the actual used polar volume (3 and 6 mL, respectively, accounting for different fractions from the final polar volume: 60 and 75 %, respectively). The reason why authors did not simply compare 50_HIGH and 80_HIGH, whose comparison would be more reliable, remains unknown. My comment is that this whole section should be rewritten.

-Section 4.4. It is not clear how the cell mass measurement was carried out. Did they authors used the weight obtained in line 364? That means it is actually dry weight not solely attributed as “cell mass”. Why did authors use 8 mL of 100 % MeOH instead of 1.5 mL of methanol and 1.5 mL of water as in the 50-HIGH conditions (i.e. the optimal condition)? Does this mean that authors did not use the same actual sample used for GC-MS to measure protein but rather seeded an extra plate for separate protein determination? Why did not authors simply dry each sample right after taking the 6 mL needed for GC-MS and then reconstitute the pellet in corresponding buffer for further BCA assay? This needs to be better discussed. Moreover, line 367 is not accurate, it is not protein extraction, it is a protein precipitation step.

-Table 2. The final conditions (50-HIGH) are the least favourable in terms of the number of metabolites (achieving the lowest number of metabolites for 10 and 20 % FCS). Of course, I am aware differences with other conditions are not very large but still this should be commented.

-Abstract, lines 36-37 and lines 290-291, about the importance on keeping ratio of protein content and cells constant. Does that mean that both need to be measured? This is in disagreement with conclusions from authors and should be further clarified.

Some other minor issues:

-Line 33, should read “its” not “it’s”.

-Line 74, can authors provide more examples in the literature in which 20 % FCS is needed?

-Fig 3 panel b, it is somehow difficult to understand how conditions 50-LOW and 80-HIGH are statistically significant but conditions 50-MEDIUM and 80-HIGH are not. Can authors double check this is correct?

-Line 128. I could not find that section 1.1

-Line 193. Do authors mean that HCT116 cells cultured with 20 % FCS showed increased RSDs, in the 50-MEDIUM condition? Then this should be added.

-Line 195. Figure 1 does not actually include RSD values. This should be modified accordingly. Also, how did authors measure the density of the pellet? Pellet from 50-HIGH seems denser than the one in 80-HIGH (see fig 1). The sentence in line 194 “80HIGH conditions the pellet was denser” needs to be rewritten.

-Line 334 reads that 80 % methanol was the optimum condition for quenching. Should not it be 50 %?

-Line 345, how long did it take to evaporate the polar phase?

-Table S6. What did authors assay in “alkane 32”? This is not explained in the text.

-Could authors provide (approximate) number of cells used?

Round 2

Reviewer 2 Report

The authors have properly addressed the comments that I raised in my former revision. Current version of the manuscript is clearer and more descriptive, thus, I can now accept the manuscript in its current form. Only few minor additional suggestions:

-Line 153: it is somehow weird that the first Table to be included is already Table 5. Tables should follow ordinal sequence starting from 1. Perhaps authors can consider including a Table with the conditions as a supporting table or simply delete reference to Tab 5 in such early stage of the text).

-Line 219: a quenching solution has the purpose of quench metabolism. If it already extracts the metabolites then the process is not quenching but quenching + extraction. Authors should consider include “first extraction step” or similar term, also in the title of section 2.2 to avoid confusion.

-Lines 397-399. Authors have clarify that under certain circumstances, intracellular protein content might be different, e.g. hypoxic vs normoxic cells. Is this statement based only on hands-on experience or is there literature that can back this up? In that case authors should consider including at least a reference.
